# Taking stock of national climate policies to evaluate implementation of the Paris Agreement

Mark Roelfsema et al.[#]

Many countries have implemented national climate policies to accomplish pledged Nationally Determined Contributions and to contribute to the temperature objectives of the Paris Agreement on climate change. In 2023, the global stocktake will assess the combined effort of countries. Here, based on a public policy database and a multi-model scenario analysis, we show that implementation of current policies leaves a median emission gap of 22.4 to 28.2 $GtCO_2eq$ by 2030 with the optimal pathways to implement the well below 2 °C and 1.5 °C Paris goals. If Nationally Determined Contributions would be fully implemented, this gap would be reduced by a third. Interestingly, the countries evaluated were found to not achieve their pledged contributions with implemented policies (implementation gap), or to have an ambition gap with optimal pathways towards well below 2 °C. This shows that all countries would need to accelerate the implementation of policies for renewable technologies, while efficiency improvements are especially important in emerging countries and fossil-fuel-dependent countries.

[#]A full list of authors and their affiliations appears at the end of the paper.

The objective of the Paris Climate Agreement is to hold average global warming to well below 2 °C above pre-industrial levels and to pursue efforts to limit the temperature increase to 1.5 °C[1]. While this objective is formulated at the global level, the success of the agreement critically depends on the implementation of climate policies at the national level. This is organised in the agreement by the requirement of countries to submit nationally determined contributions (NDCs). Countries are expected to update their NDCs in 2020. While NDCs should be submitted by every country and updated every five years, their policies and targets are not legally binding. Previous studies have highlighted that taken together, the NDCs and national policies fall significantly short of the overall ambition of the Paris Agreement[2–4]. To achieve the targets from the NDCs, countries are implementing policies at the national level. The Paris Agreement facilitates a global stocktake in 2023, which is expected to take stock of the collective efforts and to inform the preparation of more ambitious NDCs. For this, clear insights are needed into the impact of current implemented national policies from individual countries. At the moment, no peer reviewed literature exists that has assessed the global and country impact of national climate policies on the basis of a comprehensive policy inventory by using a suite of integrated assessment models, and using this to guide additional policy implementation. Such a multi-model approach using a range of model types (simulation/optimisation, general or partial equilibrium) adds to the robustness of the assessment.

The aim of this article is to fill this knowledge gap and to provide insights into the impact of national policies in comparison to emission pathways consistent with the NDCs and overall goals of the Paris Agreement. Consequently, we divide the total emissions gap between national policies and well below 2 °C pathways into an implementation gap referring to the difference between the impact of national policies and the NDCs, and an ambition gap referring to the difference between the impact of the NDCs and well below 2 °C emission pathways. The results are presented for seven large economies and the world. The analysis was done by first establishing a list of high-impact policies[5] for each G20 economy selected from a detailed open-access policy database[6], and translating these to input parameters for integrated assessment models. Subsequently, the model results allowed to assess the direct impact of these policies, as well as their interactions. The results are also presented in terms of the Kaya identity allowing to indicate how to close the implementation and ambition gaps[7,8]. The nine integrated assessment models (see Methods) used in this study have all submitted data for the 1.5 °C scenarios to the IPCC 1.5 °C report[9]. To evaluate the coherence of the national pathways, we compared the aggregated results of the integrated assessment models with similar runs of national models for the same countries.

Model-based scenarios have played a major role in supporting international climate policy already for a few decades. The focus of model analyses, however, has been mostly on exploring cost-optimal response strategies required to meet the climate temperature goals and simplified representations of national policies, typically incorporating them as overall emission reduction targets implemented via carbon prices[10–12]. The new phase of climate policy after Paris requires new information on the long-term contribution of specific policies. While some assessments have accounted for more explicit climate policy formulations in different parts of the world, these are typically single model exercises or focus only on the NDCs[11,13–16]. As such, the current work adds to the literature.

Owing to the aggregation level of most IAMs, our analysis is limited to the national policies and NDCs for G20 economies that represent 75% of total 2010 greenhouse gas emissions. It is estimated that the countries with high-impact policies, but not included in our assessment, represent around 5% of global 2010 emissions (see Supplementary Table 1). The collected policies have been made available in an open-access database[6] and cover implemented and planned national policies up to 2017. As introduction of new policies mostly occurs simultaneously with key international accords[17], this inventory contains most of the relevant policies that were introduced around the Paris Agreement. A selection from this database was made that consisted of around ten policies for each G20 country that were expected to have high impact on greenhouse gas emissions based on literature or national expert opinion, that were adopted by national governments trough legislation or executive orders, and no evidence exists of large barriers to implementation. The results are presented at the global level and for the seven large emitting economies for which national models were available, i.e., Brazil, China, the European Union, India, Japan, the Russian Federation and the United States, together representing around 65% of global 2010 greenhouse gas emissions[18].

The results show that if no additional action is taken beyond current implemented national climate policies, greenhouse gas emissions are projected to increase substantially between 2015 and 2030, although 5.3% lower compared to the hypothetical situation if these policies would not have been implemented. Current national policies together, leave a median global total emissions gap by 2030 of 22.4 Gigaton $CO_2$ equivalent (Gt$CO_2$eq) with a cost-optimal 2 °C emission pathway, and 28.2 Gt$CO_2$eq with a 1.5 °C pathway. The 2 °C global emissions gap can be reduced by a third, if conditional NDCs were fully implemented, which would close the global implementation gap, but would still leave a significant ambition gap. For seven large individual countries (China, the United States, India, the European Union, Japan, Brazil and the Russian Federation), policy implementation is expected to reduce emissions at the national level by 0 to 9% (median estimates) compared to the hypothetical situation if no policies would be implemented. This leaves a small implementation gap for China, India, Japan, Russian Federation as they are close to achieving their NDC, while this is not the case for the European Union, United States and Brazil, but their ambition gap is smaller as NDCs are close to the cost-optimal 2 °C pathways.

## Results

**Global implementation and total emissions gap**. In total, five scenarios were evaluated (see Table 1 and Supplementary Note 1). The starting point of all scenarios is the SSP2 scenario[19,20], which is a middle-of-the-road scenario assuming a business-as-usual conduct representing no new climate policies implementation after 2010 (no new policies scenario). The national policies scenario represents the impact of policies implemented domestically to fulfil the NDC promises that are included in the NDC scenario. The 2 °C and 1.5 °C scenarios look into cost-optimal implementation of the overall goals of the Paris Agreement. To provide guidance on enhancing policy implementation, the impact of policies is decomposed by computing a set of indicators based on the Kaya identity (see Supplementary Note 2). Besides greenhouse gas emissions, also the share of low-carbon (no fossil-fuels without carbon capture and storage) technologies and energy efficiency is presented.

Under the No new policies scenario, the models project an increase in global greenhouse gas emissions to 63.9 Gt$CO_2$eq (61.0–69.1; median and 10th to 90th percentile range over all model results) by 2030. This is mostly driven by an increase in emissions related to transport, industry and power production in developing countries, but still to lower per-capita levels than developed countries. Implementation of national policies is not

| Table 1 Main assumptions on climate policy implementation per scenario. | | |
|---|---|---|
| **Scenario** | **Policy assumptions** | |
| | **Until 2030** | **After 2030** |
| No new policies | None | None |
| National policies | Implementation of current domestic policies | Equivalent effort to policy implementation before 2030 |
| NDCs | Full implementation of conditional national NDCs | Equivalent effort to NDC implementation before 2030 |
| 2 °C/1.5 °C | Each country implements current implemented polices until 2020 and starts with cost-effective implementation to achieve the 2 °C/1.5 °C target between 2020 and 2030 with high (>66%) probability, thereby staying within a global carbon budget of 1000 $GtCO_2$ and 400 $GtCO_2$ in the 2011–2100 period | Continuation of cost-effective implementation to achieve the 2 °C/1.5 °C target |

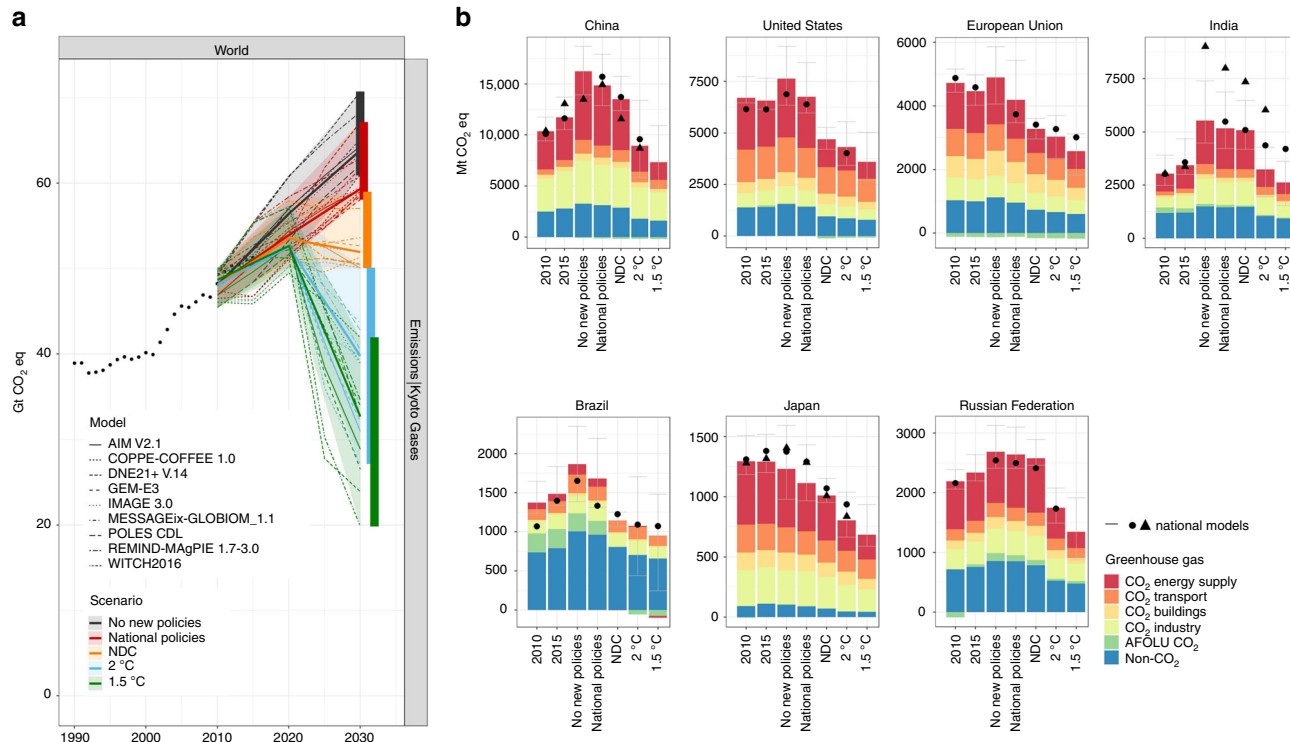

**Fig. 1 Greenhouse gas emissions on a global level and seven large countries under different scenarios. a** Global greenhouse gas emissions for total greenhouse gases (in $GtCO_2eq$) and nine integrated assessment models between 2010 and 2030. **b** Average greenhouse gas emissions (in $MtCO_2eq$) of all models by 2010, 2015 and 2030 for $CO_2$ emissions per sector and total non-$CO_2$ emissions (blue), including the 10th–90th percentile ranges for total greenhouse gas emissions of the multi-model ensemble (error bars). $CO_2$ emissions have been separated into those related to energy supply (red), transport (dark orange), buildings (light orange), industry (yellow) and AFOLU (agriculture, afforestation, forestry and land-use change) (green). National models are China-TIMES and IPAC for China, GCAM-USA for the United States, PRIMES for the EU, AIM India and India MARKAL for India, RU-TIMES for the Russian Federation, BLUES for Brazil and AIM/Enduse and DNE21 + for Japan. For both panels, $CO_2$ equivalent greenhouse gases have been calculated using the 100-year Global Warming Potential from the IPCC Fourth Assessment Report. The data is available in the source data.

projected to reverse the increase of global emissions by 2030, and would result in emission levels of 59.3 $GtCO_2eq$ (58.4–63.7) (Fig. 1), which is a 5.3% (3.8%–7.9%) reduction relative to the No new policies scenario (see Table 2). However, it covers 15.4% (10.8%–19.0%) of the emissions gap between No new policies and the 2 °C pathway by 2030, and this is 11% (7.6%–15.9%) for the 1.5 °C pathway.

Although the global low-carbon share of final energy under the National Policies scenario increases by 1 percentage point (1 pp) to 14.3% (9.3%–19.8%) by 2030, and the energy intensity improves by 20.5% (16.1%–24.7%) between 2015 and 2030, final energy use still increases (see Fig. 2). Most emission reductions

under the National policies scenario are induced by high-impact policies that target $CO_2$ emissions (Fig. 1). Furthermore, 45% (30–70%) of the emission reductions are projected to come from countries that are member of the the Organisation for Economic Co-operation and Development (OECD).

For achieving conditional NDCs, deeper reductions are necessary than those achieved by national policies only. The implementation of conditional NDCs (NDC scenario) is projected to result in 51.9 (50.4–57.4) $GtCO_2eq$ greenhouse gas emissions by 2030, a low-carbon share of final energy at 16.8% (12.6%–25.2%), and 23.5% (17.9%–30.0%) in energy-intensity improvement between 2015 and 2030. This means that national

**Table 2 Absolute (GtCO$_2$eq) and percentage impact of policy implementation relative to no new policies scenario, and implementation gap with NDC scenario for the world, China, United States, India, EU, Japan, Brazil and Russian Federation (median value and 10–90% in brackets).**

| Economy | Absolute impact of policy implementation relative to no new policies scenario (GtCO$_2$eq) | Percentage impact of policy implementation relative to no new policies scenario (%) | Absolute reductions between national policies and conditional NDCs (GtCO$_2$eq) | Percentage reductions between national policies and conditional NDCs (%) |
|---|---|---|---|---|
| World | 3.5 (2.3, 5.2) | 5 (4, 8) | 7.7 (5.3, 9.7) | 13 (9, 16) |
| China | 0.7 (0.5, 2.3) | 5 (2, 14) | 0.9 (−0.5, 3.7) | 6 (−3, 22) |
| United States | 0.4 (0.3, 1.2) | 6 (4, 13) | 2.1 (1.5, 3.2) | 31 (22, 38) |
| European Union | 0.5 (0.3, 0.6) | 9 (7, 15) | 0.7 (0.6, 1.8) | 19 (15, 33) |
| India | 0.1 (0, 0.5) | 3 (0, 7) | 0.1 (−0.1, 0.3) | 2 (−3, 6) |
| Japan | 0.1 (0, 0.1) | 7 (2, 8) | 0 (0, 0.3) | 4 (−4, 23) |
| Brazil | 0.0 (0, 0.2) | 3 (0, 11) | 0.5 (0.2, 1) | 30 (14, 44) |
| Russian Federation | 0.0 (0, 0) | 0 (0, 2) | 0.1 (−0.1, 0.2) | 3 (−3, 7) |

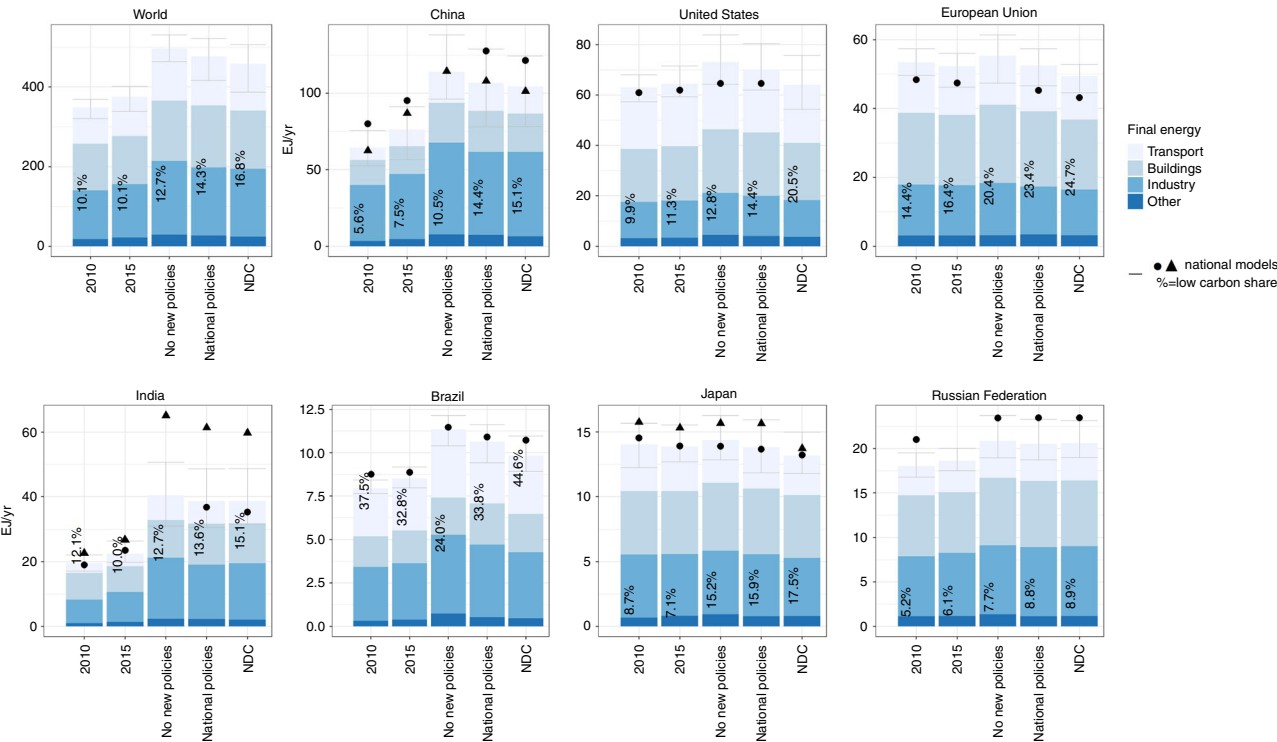

**Fig. 2 Final energy and the low-carbon share of final energy on the global level and seven large countries under different scenarios.** Average total final energy for 2010, 2015 and 2030 of nine global integrated assessment models is subdivided into sectors: transport, buildings, industry and other. Total final energy includes the 10th to 90th percentile ranges for total final energy (error bars). The black dots/triangles indicate final energy based on national model estimates (China-TIMES and IPAC for China, GCAM-USA for the United States, PRIMES for the European Union, AIM India and India MARKAL for India, RU-TIMES for the Russian Federation, BLUES for Brazil and AIM/Enduse and DNE21+ for Japan). The data is available in the source data.

policies together leave a significant global implementation gap with respect to the NDC targets by 2030, which is 7.7 (5.3–9.7) GtCO$_2$eq for emissions (see Table 3). This gap by 2030 can be closed by increasing the low-carbon share by 2.8 pp (1.5–4.7 pp), and decreasing energy intensity by 12.7% (9.1%–16.1%). Final energy reductions under the NDC scenario compared with the national policies scenario, occur especially in the transport and buildings sector (see Fig. 2).

**Uncertainty range.** The different integrated assessment models provide a range of outcomes for changes in greenhouse gas emissions due to policy implementation between 2015 and 2030. This range is a result of the differences in historical emissions[21],

different assumptions about socio-economic growth rates, different impact of policy implementation in models, and finally real uncertainty as a result of structural model differences (see Methods). The differences in historical emissions are in line with estimates of uncertainty in historical emission inventories (10% in total greenhouse gas emissions)[22], but it clearly translates into a contribution to uncertainty for 2030. In addition, an estimate of the contribution of socio-economic factors can be obtained by comparing the 2015 and 2030 emission range under the No new policies scenario. This shows a 2030 range that is 50% larger than the 2015 range. The different impact of policies implemented in models has been estimated by considering the impact of all policies implemented in the models and estimating those that

**Table 3 Absolute (GtCO$_2$eq) and percentage emissions gaps by 2030, on the global level and for China, the United States, the European Union, India, Japan, the Russian Federation and Brazil.**

| | Absolute emissions gap between national policies and 2 °C scenarios | Absolute emissions gap between national policies and 2 °C scenarios | Emissions gap in percentages between national policies and 1.5 °C scenarios | Emissions gap in percentages between national policies and 2 °C scenarios |
|---|---|---|---|---|
| World | 22.4 (13.6, 29.6) | 36 (23, 49) | 28.2 (19.8, 42.2) | 45 (33, 65) |
| China | 5.9 (4.2, 8.4) | 41 (24, 59) | 7.2 (5.3, 11) | 53 (33, 66) |
| United States | 2.3 (1.5, 3.9) | 37 (24, 47) | 2.9 (2.2, 5) | 43 (33, 66) |
| European Union | 1.6 (0.6, 1.9) | 31 (14, 43) | 1.4 (0.9, 3.1) | 33 (25, 65) |
| India | 2.1 (1.1, 2.7) | 33 (21, 54) | 2.6 (1.6, 3.2) | 45 (34, 63) |
| Japan | 0.4 (0.1, 0.5) | 25 (14, 40) | 0.5 (0.3, 0.6) | 37 (28, 47) |
| Brazil | 0.7 (0.4, 1) | 40 (20, 70) | 0.9 (0.4, 1.2) | 54 (23, 83) |
| Russian Federation | 0.9 (0.5, 1.2) | 34 (23, 43) | 1.3 (0.7, 1.9) | 49 (26, 68) |

were not included based on the IMAGE model results (see Methods). Based on this analysis, it can be concluded that assumptions on socio-economic factors explain the largest part of the ranges in the results for 2030; while the differences in policy impact explain about 1/3 of them.

**Impact of national policies for seven large G20 economies**. The scenarios allow for evaluation of climate policy at the national level (although obviously limited by model detail). Policy implementation is estimated to result in reductions of 0% (0–2%) for the Russian Federation to 10% (4–12%) for the United States, relative to the no new policies scenario (see Table 2). The largest absolute emission reductions under the National policies scenario occur in the CO$_2$ energy supply and transport sector, in all countries, except for Brazil, where reductions also occur in the AFOLU sector (although AFOLU emission estimates are inherently uncertain, already for historical estimates[23]). The largest percentage of reductions is projected in the transport sector for the United States and India, the industrial sector for the EU, and the energy supply sector for China and Japan. In the Russian Federation, the National policies scenario hardly triggers emission reductions, compared to the no new policies scenario.

Implementation of national policies still leaves an implementation gap with NDCs of 3% (3–7%) for the Russian Federation to 28% (22–37%) for the United States (see Table 2). With national policies until cut-off date before 2017, China, India, Japan and Russian Federation are projected to come close to achieving their NDC targets with national policies by 2030. In Brazil, the European Union, and the United States, the median estimate of the National policy scenario is further removed from the NDC level. Note that very recent policy updates since 2017, or planned policies in the pipeline to be implemented were not included. We have compared the results of the global models also to the outcomes of the same scenarios from national models from each individual country. These results confirm the above trends, although the absolute levels differ in a few cases (Figs. 1 and 2)

**Global emissions gap and for seven large G20 countries**. In order to implement the objectives of the Paris Agreement, all national policies together should reduce emissions enough to keep global warming below the 2 °C and 1.5 °C temperature limits. We evaluate this by comparing the results of the policy scenarios with cost-optimal scenarios for these temperature targets. This shows a total emissions gap between the National policies scenario and the cost-optimal scenarios in 2030 of 22.4 GtCO$_2$eq (13.6–29.6) for the 2 °C limit (high probability), and 28.2 GtCO$_2$eq (19.8–42.2) for the 1.5 °C limit (see Table 3 and Fig. 3). This is respectively a global reduction of 36% (23–49%)

and 45% (33–65%) by 2030 relative to the national policies scenario.

The Kaya identity allows to break this up into an energy mix gap (share of low-carbon emitting technologies in final energy) and an efficiency gap (final energy-intensity improvement relative to the results of the implementation of national policies), and a carbon-intensity gap (see Supplementary Figs. 1 and 2). To close the gap by 2030 with the National policies scenario, the non-fossil share would need to increase by 6.9 pp (4.0%–12.3%) (energy mix gap), and the energy-intensity needs to improve by 9.6% (4.8%–24.7%)) (efficiency improvement gap). These numbers are 13.0% (7.2%–24.0%) and 17.5% (12.5%–26.8%) for the 1.5 °C case (see Fig. 3). Global annual mitigation costs per GDP by 2030, under the national policies scenario, are small, and increase to 0.9% (0.3%–2.2%) under the 2 °C scenario, and to 1.3% (1.0%–4.0%) under the 1.5 °C scenario (see Fig. 3). The global emissions gap with the 2 °C scenario can be reduced by a third, if conditional NDCs would be fully implemented, leaving a median ambition gap of 16.5 GtCO$_2$eq (6.4–21.0) with 2 °C pathways and 21.2 GtCO$_2$eq (12.2–31.6) with 1.5 °C pathways.

For the seven individual G20 countries, greenhouse gas emissions by 2030 would need to decrease compared to the national policies scenario by 25 to 41% (median) to stay on track to keep temperature below 2 °C, while this is 33 to 54% (median) under the 1.5 °C scenario (see Table 2 and Fig. 3). These gaps can be closed by strongly increasing the low-carbon share of final energy by 5.4 pp for the European Union to 8.5 pp for China to stay below 2 °C, and between 5.4 pp in the European Union to 20.2 pp in China for the 1.5 °C case. Projections for final energy intensity give a different picture, where the difference between the National policies scenario and the 2 °C scenarios are small for the European Union, Japan and the United States, somewhat larger (and more uncertain) for Brazil, and largest for China, India and the Russian Federation (See Fig. 3). Closing the gap between national policies and 2 °C or 1.5 °C pathways by 2030 would result in additional median mitigation costs per GDP of between 0.5% for the European Union to 2.8% for the Russian Federation for the 2 °C case, while this is 0.6% to 3.4% for the 1.5 °C case (see Fig. 3).

**Mid-century impact of national policies**. To give an indication of the short-term impact of national policies in the context of the long-term global targets, we present the indicator that is defined as the cumulative emissions in the 2011–2050 period divided by the 2010 emissions, and in addition assume countries pursue the same national efforts between 2030 and 2050 under the National policies scenario by keeping total percentage emission reductions relative to the No new policies scenario constant. The indicator allows for comparing countries with different absolute emission

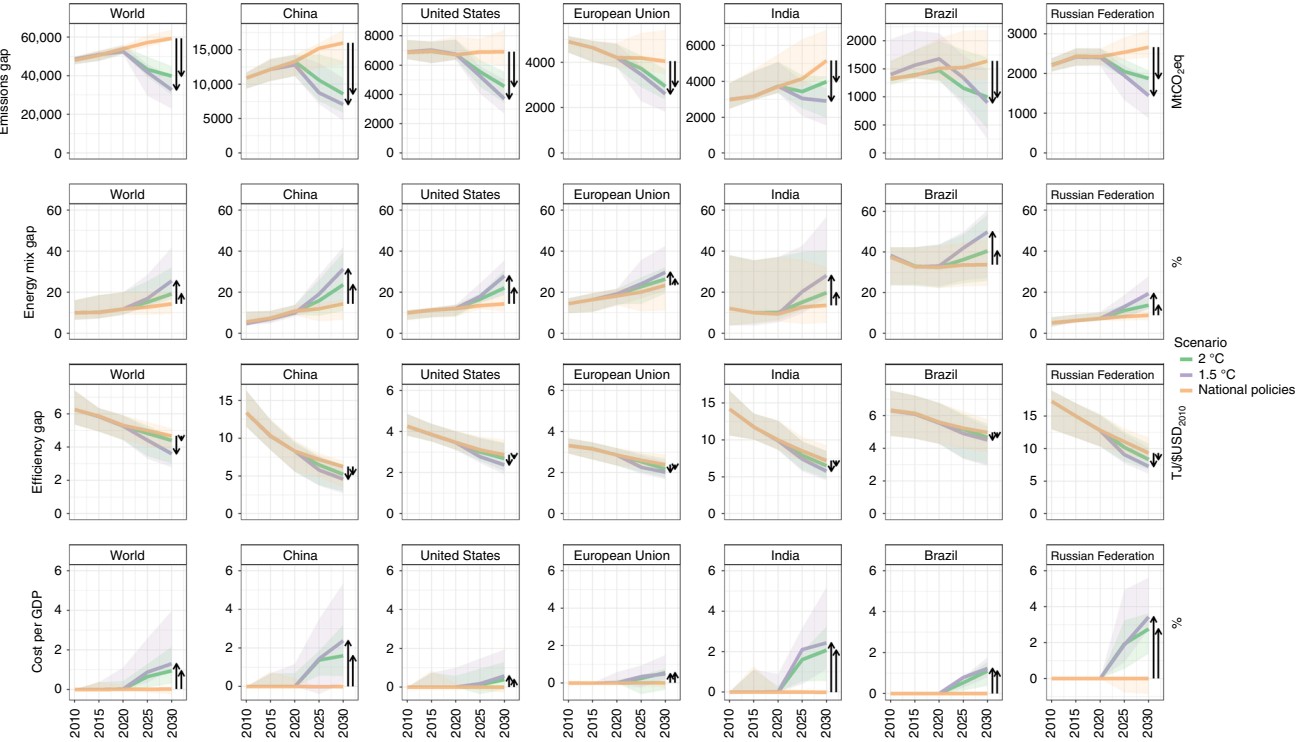

**Fig. 3 Indicators derived from Kaya identity and costs per GDP between 2010 and 2030 on a global level and for seven large countries under different scenarios.** The median (lines) and 10th–90th percentile ranges (areas) from nine integrated global assessment models on emissions, energy mix and efficiency gaps and mitigation costs per GDP. These gaps are represented by total greenhouse gas emissions (MtCO$_2$eq), low-carbon share of final energy (%), final energy intensity in GDP (TJ/USD$_{2010}$) and total mitigation costs per GDP (%) between national policies and well below 2 °C scenarios. The data is available in the source data.

levels, and provides the number of years an economy can emit at 2010 emission levels while staying below the total cumulative emissions of the next 40 years. A value of 40 indicates that, on average, the emission level will remain constant. In the same way as for the shorter period until 2030, comparison of the results with the trajectories for the 2 °C and 1.5 °C maximum temperature increases shows a large gap (Fig. 4). Interestingly, the NDC projections by 2050 for the European Union, Brazil, and the United States are relatively close to the 2 °C scenario, suggesting that these regions would mostly need to ensure that their national policies more closely lead to the NDC target (which may possibly already be achieved through very recent policy updates). It should, however, be noted that cost-optimal implementation (equal marginal costs in all regions) leads to higher costs, as a percentage of GDP, in low-income regions and, therefore, is a fair way to implement the Paris Agreement (see Supplementary Note 3) only if complemented by financial transfers. Effort-sharing approaches based on equity considerations tend to suggest larger reduction targets for high-income regions[24].

The 2 °C and 1.5 °C model ranges for Brazil are large as a result of the uncertainty in land-use-related emissions. In terms of cost-optimal mitigation, large reductions in each G20 economy are necessary to stay within the 400 Gt carbon budget. The median estimate for cumulative emissions relative to 2010, under this scenario, is at a similar level, between 20 and 25, except for Brazil and India, indicating that given the estimated cumulative emissions in the national policies scenario, strong efforts are essential by almost all countries.

## Discussion
The results show that for all countries there is either a significant implementation gap or ambition gap. Unless governments

increase ambition, the collective effort of current national policies significantly stays short of the objectives of the Paris Agreement and even fails to meet the joint ambition secured in NDCs. The results have strong implications beyond 2030. Previous literature has shown that inadequate near-term reduction efforts imply that a substantially higher rate of transformation will be needed to comply with the 2 °C limit[11], stranded assets[25] and substantially higher mitigation costs in the long term, and reduced techno-economic mitigation potential due to carbon lock-in[26].

In all, 2 °C and 1.5 °C pathways in this study are calculated assuming cost-optimal implementation, but it might not be the most realistic approach to deriving national reduction targets, as it would typically lead to relatively high costs in low-income countries. In contrast, effort-sharing approaches based on equity principles would lead to lower allowance of cumulative emissions in the EU, Japan, the Russian Federation and the United States, and to higher allowances for India (see Supplementary Fig. 3), resulting in an opposite impact on the gap between national policies and these allowances. If cost-effective climate policy would be adopted, emission trading or transnational climate financing could still ensure a cost-optimal implementation. If less cooperation between countries is assumed, a different allocation would increase total costs of implementation.

One crucial question that arises from this analysis is how to speed up implementation to achieve NDCs, and increase ambition to stay on track to meet well below 2° goals? The current policy implementation is weak and includes significant gaps (e.g., industry, freight transport policies). Moreover, it is also often fragmented in terms of the use of policy instruments and the coverage of sectors and countries. A redesign of current policy mixes consisting of more coherent policies, including for instance the use of economy-wide financial instruments[27], may respond to

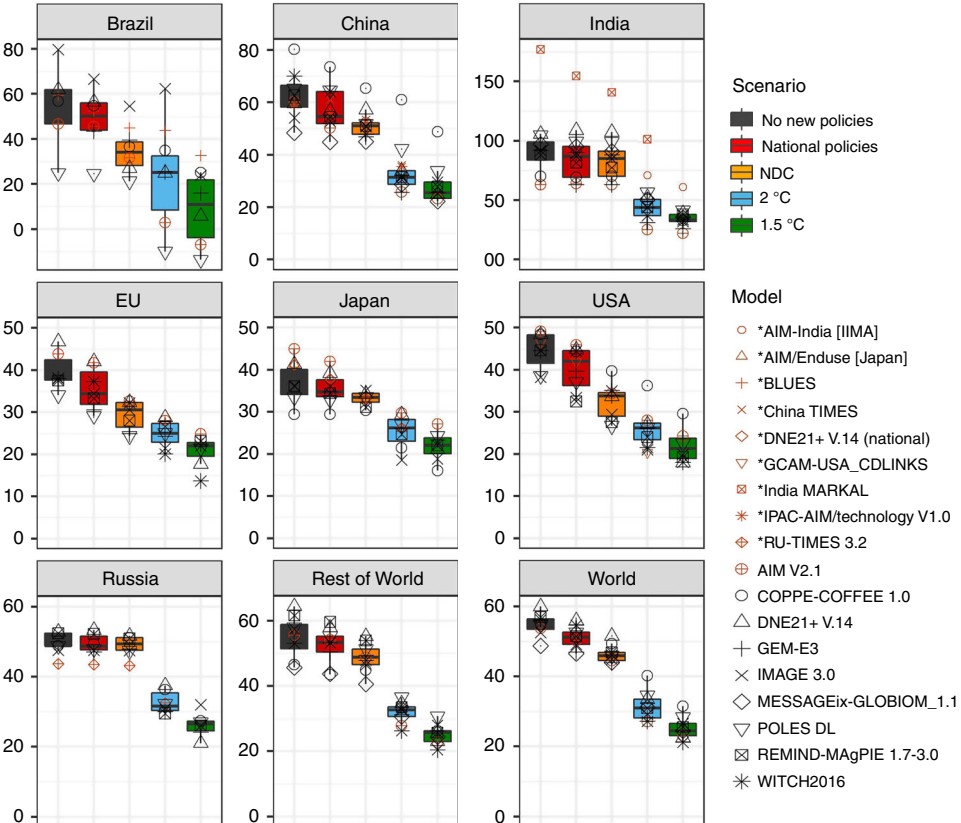

**Fig. 4 Cumulative CO$_2$ emissions in the period 2011–2050 period relative to 2010 emissions on the global level and for seven large countries under different scenarios.** The box plots indicate the median, 25th to 75th percentile range, while the black data points show the full global model range. The brown coloured markers indicate the results from the national models. The data is available in the source data.

the current call for strengthened policies. In practical terms, it is possible to draw lessons from the policy mixes used in our analysis—for instance by identifying to most successful mitigation measures. In identifying such good practices it is important to evaluate measures in terms of cost effectiveness but also in terms of reducing public policy constraints such as distribution of costs[28], ability to address uncertainty[29], and political feasibility to intervene in the economy[30]. A careful redesign in combination with international cooperation could avoid carbon leakage to other sectors and countries, avoid stranded assets[31], and increase regulatory power of governments.

In 2020, countries are expected to submit updated NDCs to the Paris Agreement. However, the global stocktake discussed in this article shows that large enhancements are necessary if we want to maintain the window to limiting temperature increase to well below 2 °C, or even pursue efforts to limit this to 1.5 °C. In order to do so, all countries would need to accelerate the implementation of renewable technologies, while efficiency improvements are especially important in emerging countries (China, India, Brazil) and fossil-fuel-dependent countries (Russian Federation). From this we conclude that the global stocktake in the Paris Agreement's process would need to go beyond presenting emission gaps, but insights and guidance for how to close this gap are important. Integrated assessment models can support the policy process. At first, the national policy scenario used in this analysis could be assessed in more detail and give insights into the impact of different individual policies. In addition, the models are well furnished to present effective mitigation options to countries for policy enhancement by giving the tradeoffs between impact and costs of different policy packages in the context of global

efforts. Other effectiveness criteria could be captured with different scenarios. Finally, as the new policy questions require more detailed information, model development could go into the direction of including more countries, sectors and actors or link to bottom-up energy and land use models.

## Methods

**Model exercise.** The assessment of the impact of national climate policies on greenhouse gas emissions is based on the model exercise that was done as part of the CD-LINKS project, and for which guidelines were described in the global and national model protocols[32,33]. This project aimed, among other things, to develop global low development pathways on a global level and for G20 economies, including an explicit representation of near-term policy trends. For this paper, we selected seven large G20 economies in terms of greenhouse gas emissions (Brazil, China, the EU, India, Japan, Russian Federation, United States), for which also national climate and energy models were available in the project.

**Integrated assessment models.** Integrated assessment models (IAMs) describe key processes in the interaction of human development and natural environment and are designed to assess the implications of achieving climate objectives[2,34]. The model exercise that assessed the impact of climate policies was done by nine IAMs that have global coverage, and ten national models that represent a specific G20 economy (see Table 4). A more detailed description of model structure and policy implementation can be found in the Supplementary Note 4, and for some models at the IAMC wiki[35]. These models differ in country and sector aggregation level, and also in the way they mimic decisions on climate policy. All models include dynamic pricing, and therefore local climate policy will result in lower implementation in other regions with less policies. However, only the economic models explicitly account for carbon leakage. In addition, as most models assume one central planner, behaviour or decisions of different actors and the role of institutions is often not explicitly taken into account. This implies that most models (especially with simple representation of the economy) have only a limited ability to reflect the specific social and economic dynamics of the developing and transition economies[36]. Some phenomena, such as the green paradox, can only be

**Table 4 Participating integrated assessment models in the model exercise to assess the impact of climate policies.**

| Model | Coverage IAM model | Institute | Model type |
|---|---|---|---|
| AIM V2.1 | Global | Kyoto University and National Institute for Environmental Studies (NIES, Japan) | Recursive dynamic, general equilibrium |
| COPPE-COFFEE 1.0 | Global/national | Energy Planning Programme, COPPE, Universidade Federal do Rio de Janeiro (COPPE, Brazil) | Perfect foresight, general equilibrium |
| DNE21 + V.14 | Global/national | Research Institute of Innovative Technology for the Earth (RITE, Japan) | Perfect foresight, partial equilibrium |
| GEM-E3 | Global/national | Institute of Communication and Computer Systems (ICCS, Greece) | Recursive dynamic, General equilibrium |
| IMAGE 3.0 | Global | PBL Netherlands Environmental Assessment Agency (PBL, The Netherlands) | Recursive dynamic, partial equilibrium |
| MESSAGEix-GLOBIOM_1.0 | Global | International Institute for Applied Systems Analysis (IIASA, Austria) | Perfect foresight, general equilibrium |
| POLES CDL | Global | Joint Research Centre (JRC, EU) | Recursive dynamic, partial equilibrium |
| REMIND-MAgPIE 1.7-3.0 | Global | Potsdam Institute for Climate Impact Research (PIK, Germany) | Perfect foresight, general equilibrium (REMIND) recursive dynamic, partial equilibrium (MAgPIE) |
| WITCH2016 | Global | Centro Euro-Mediterraneo sui Cambiamenti Climatici (CMCC, Italy) | Perfect foresight, general equilibrium |
| AIM/Enduse[Japan] | National | Kyoto University and National Institute for Environmental Studies (NIES, Japan) | Recursive dynamic, partial equilibrium |
| AIM India [IIMA] | National | Indian Institute of Management (IIM, India) | Recursive dynamic, general equilibrium |
| BLUES | National | Energy Planning Programme, COPPE, Universidade Federal do Rio de Janeiro (COPPE, Brazil) | Perfect foresight, partial equilibrium |
| China TIMES | National | Tsinghua University (TU, China) | Recursive dynamic, partial equilibrium |
| GCAM-USA_CDLINKS | National | Pacific Northwest National Laboratory (PNNL, United States) | Recursive dynamic, partial equilibrium |
| India MARKAL | National | The Energy Resources Institute (TERI, India) | Dynamic least cost optimisation |
| IPAC-AIM/technology V1.0 | National | National Development and Reform Commission Energy Research Institute (NDRC-ERI, China) | Recursive dynamic, general equilibrium |
| PRIMES_V1 | ICCS | Institute of Communication and Computer Systems (ICCS, Greece) | Perfect foresight, partial equilibrium |
| RU-TIMES 3.2 | National | National Research University-Higher School of Economics (HSE, Russian Federation) | Perfect foresight, partial equilibrium |

**Table 5 Number of high-impact policies selected for implementation in the IAM models, per sector and country (details in, Supplementary Table 3).**

| Sector | Brazil | China | European Union | India | Japan | Russian Federation | United States of America | Other G20 countries | Total |
|---|---|---|---|---|---|---|---|---|---|
| Economy-wide | 3 | 9 | 11 | 0 | 3 | 1 | 1 | 11 | 39 |
| Energy supply | 6 | 10 | 0 | 9 | 7 | 6 | 3 | 37 | 78 |
| Transport | 5 | 10 | 2 | 9 | 2 | 0 | 5 | 20 | 53 |
| Buildings | 1 | 1 | 2 | 0 | 1 | 1 | 6 | 4 | 16 |
| Industry | 0 | 3 | 0 | 4 | 1 | 0 | 0 | 1 | 9 |
| AFOLU | 4 | 3 | 0 | 2 | 2 | 0 | 1 | 8 | 20 |
| Total | 19 | 36 | 15 | 24 | 16 | 8 | 16 | 81 | 215 |

represented by most models in an explicit scenario design. However, the models with less economic detail often have a more detailed representation of technologies in different sectors enabling them to take into account technological learning.

**Selection and model implementation of policies.** Climate policy on the national level, in this research, is defined as the result of climate policy formulation and climate policy implementation that encompasses aspirational goals not secured by legislation, national targets that are secured by legislation, and policy instruments designed to implement these targets. Only implemented policies were included in this analysis, and are defined as policies adopted by the government through legislation or executive orders, and non-binding targets backed by effective policy instruments.

First, climate policies were collected with the help of national experts and a literature study (see Supplementary Table 2), and were stored in an open-access database[6]. With the help of national experts, a selection of high-impact policies was made and translated into model input indicators[5]. This inventory includes climate

and energy policies for the G20 economies, and details the instruments, targets and sectors (see Table 5 and Source Data). It was evaluated with and expanded by national experts in two rounds. The cut-off data for the selection of policies was 31 December 2016, and it should be noted that the policy environment is constantly changing. Two policy changes with a possibly high impact have occurred since this date: the United States is not likely to implement the 2025 standards for light-duty vehicles, although current standards are implemented until 2021 (The Clean Power plan, already was not included in the list of high-impact policies), and the European Union adopted a comprehensive set of climate actions that goes beyond the policies that we included in our analysis. In addition, although the United States announced its withdrawal of the Paris Agreement, this would only enter into effect by November 2020.

Policy instruments were represented in the integrated assessment as explicit as possible, but simplification was sometimes necessary, thereby considering replicating the impact on greenhouse gas emissions and energy as most important. In practice, policy instruments are implemented to achieve national, often aspirational goals (not secured by legislation or executive orders). These

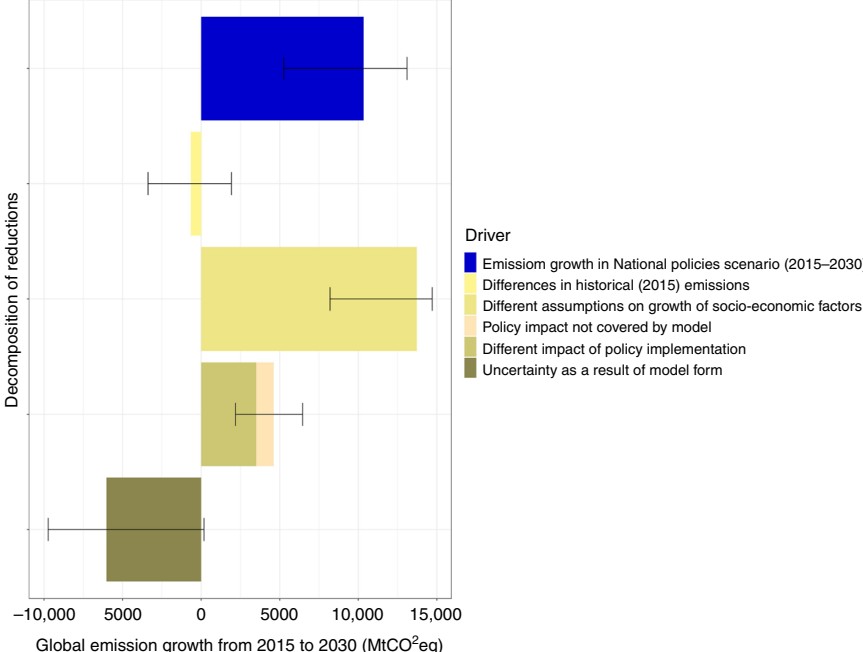

**Fig. 5 Decomposition of total median emission growth between 2015 and 2030 under National policies scenario, error bars range between 10th to and 90th percentiles.** The data is available in the source data.

aspirational goals are documented in national policy documents (e.g., National Communication, strategy documents). In some cases, we could directly implement policy instruments in IAMs, such as carbon taxes or regulations (e.g., vehicle fuel-efficiency standards). In other cases, we included aspirational policy targets to represent currently implemented policies, but only if they were backed by effective policy instruments. This was for example the case with feed-in tariffs or renewable auctions. If the policy instrument would end before the policy target year, we assumed continuation of this instrument until the target year of the aspirational goal. In case a G20 country is part of a larger model region, the policy (indicator) is aggregated by assuming business-as-usual for those countries without policies, and implementation of the policy for countries with policies[32]. In some cases, models with less sector detail used indicators (such as $CO_2$ or final energy reduction) based on the impact of policies from more detailed models or on literature. See the Supplementary Note 4 and Supplementary Table 4 for information on how policies were implemented for each global model.

**Scenarios.** The starting point for the scenario design was the ADVANCE project[10]. The National policies scenario corresponds to the inventory that contains energy and climate policies implemented in G20 economies[5]. Between 42 and 94% of the high-impact policies from the seven G20 economies were implemented in the nine IAMs considered in this paper, and are estimated to represent 50 to 100% of possible greenhouse gas reductions (see Supplementary Note 2 and Supplementary Table 5). Note that global results also include G20 policies for Argentina, Australia, Canada, Indonesia, Mexico, Republic of Korea, Saudi Arabia and South Africa, which were not individually addressed in this paper. The national policies were implemented for the period from 2010 to 2030, and equivalent effort was assumed after 2030. This was defined as a constant percentage reduction relative to the No new policies scenario or similar forms of continued ambition. The NDC scenario was based on information from the NDCs on greenhouse gas reduction targets energy and land-use policies and on additional information from Kitous et al.[37], den Elzen et al.[38], Grassi et al.[39] (land use estimates), and information from the UNFCCC (see Supplementary Tables 6–8 for details). The NDC targets can be divided into absolute emission reduction targets, business as usual reductions, emission-intensity reductions, and projects absent of greenhouse gas emission targets[40]. All G20 countries' NDCs are of the first three types. In general, NDC targets for G20 economies are defined for the year 2030, but the US NDC target is defined for the year 2025. The NDCs for China and India are represented by greenhouse gas intensity targets, renewable targets and forestry measures, which could not be translated into one specific absolute greenhouse gas emission level. The 2 °C scenario assumes implementation of national policies until 2020 and cost-optimal mitigation measures after 2020, to stay within the carbon budget of 1000 $GtCO_2$ between 2011 and 2100. This is in line with the carbon budgets of 590 to 1240 $GtCO_2$ from 2015 onwards, which would limit global warming by 2100 to below 2 °C, relative to pre-industrial levels with at least 66% probability. The 1.5 °C scenario starts with cost-optimal deep mitigation measures after 2020, and explores the efforts necessary to keep global warming below 1.5 °C by 2100, with about 66% probability, keeping cumulative carbon emissions within 400 $GtCO_2$ between 2011

and 2100. Both budget assumptions are based on the ADVANCE project[10], and in line with the estimate for 66% probability from Table 2.2 from the IPCC AR5 Synthesis report[41].

**Indicators to track progress.** To give insights into policy impact, we have used a variant of the framework of tracking indicators related to the Paris Agreement[7,8] (see Formula 1.1–1.3). $CO_2$ per GDP can be decomposed into energy intensity (final energy/GDP), low-carbon share of final energy (%), and utilisation rate ($CO_2$/fossil energy). The most pronounced differences between countries and scenarios for these indicators are visible for the low-carbon share of final energy (%) and energy intensity (final energy/GDP) (results are shown in Supplementary Figs. 1 and 2), and are discussed in the article. However, not only was the impact of policies on $CO_2$ emissions analysed, but also total greenhouse gas emissions and individual greenhouse gas emissions ($CO_2$ energy, $CO_2$ industrial processes, $CO_2$ AFOLU, non-$CO_2$). In addition, we have added mitigation costs per GDP to assess the affordability of climate policy implementation. Partial equilibrium models such as IMAGE and POLES report these costs in terms of area under the MAC curve (e.g., direct mitigation costs), while equilibrium models such as MESSAGE, REMIND and WITCH report in terms of consumption losses. MAC cost measures tend to exclude existing distortions in the economy[42]. But as GDP is an exogenous variable in partial equilibrium models, consumption loss is not available.

The Kaya decomposition is

$$CO_2 = POP * \frac{GDP}{POP} * \frac{CO_2}{GDP} \tag{1.1}$$

$$\frac{CO_2}{GDP} = \frac{TPES}{GDP} * \frac{CO_2}{TPES} = \frac{TPES}{GDP} * \frac{FE}{TPES} * \frac{CO_2}{FE} = \frac{FE}{GDP} * \frac{CO_2}{FE} \tag{1.2}$$

$$\frac{CO_2}{GDP} = \frac{FE}{GDP} * \frac{FE_{fossil}}{FE} * \frac{CO_2}{FE_{fossil}} = \frac{FE}{GDP} * \left(1 - \frac{FE_{non-fossil}}{FE}\right) * \frac{CO_2}{FE_{fossil}} \tag{1.3}$$

where
POP population, GDP gross domestic product, TPES primary energy, FE final energy

**Results.** The results are presented (unless otherwise stated) using the median estimate of all model results, and in addition presenting the 10th and 90th percentiles of these ranges. Differences in greenhouse gas emissions between scenarios (e.g., implementation gap and total emissions gaps) are calculated by first taking the difference per model and then determining the median and percentiles of the range of differences.

The results from this study show that, for national policies, greenhouse gas emissions by 2030 would be somewhat higher and for well below 2 °C scenarios lower than earlier studies indicated[2,4,42] (which were based on only one model or had less detail on national policy implementation) (see Supplementary Figs. 4–7).

**Uncertainty**. Emission growth under the National policies scenario by 2030 (See Fig. 1) can be decomposed into five drivers that, together, represent the total impact (blue bar in Fig. 5). First, historical calibration, which is calculated as the difference between 2015 model emissions and the PRIMAP[26] (version 1.2) data set. Second, socio-economic growth assumptions, calculated as the emission growth between 2015 and 2030 under the No policy scenario. Third, policy impact on greenhouse gas emissions, calculated as the difference between 2030 emissions under the No policy scenario and the National policies scenario, including an estimate for the emission reductions for those policies (see Supplementary Table 5 and 9) that could not be implemented in certain models (See Supplementary Note 2). Fourth, real uncertainty represented by model form and heterogeneity.

This shows that the impact of historical calibration on the projected global growth in emissions between 2015 and 2030 is small; this growth is much more dependent on socio-economic factors such as GDP and population growth. Of the total impact, the policy impact is around one third, and a somewhat larger part is real uncertainty.

**Effort-sharing**. The 2 °C and 1.5 °C scenarios assume cost-optimal implementation of the reduction measures after 2020 with the lowest overall mitigation costs. The result is implementation of measures in countries where this is cheapest, but this does not imply that the implementing country would need to face all the costs. These costs can be shared, and thus financed by other countries. The financial flows could be calculated if emission allowances per country are based on so-called effort-sharing approaches representing different equity principles[24,43,44], for example, categorise the effort-sharing approaches in the literature based on the four basic equity principles, i.e., responsibility, equality, capability and cost effectiveness, and present the regional greenhouse gas emission allowances in 2020, 2030 and 2050 for these categories. The equity principles were also applied to the carbon budgets (cumulative emissions) for both the 2011–2050 and the 2011–2100 period[24], based on calculations from the FAIR model[45], see the Supplementary Fig. 3, for comparison with the results from our study.

**Model result adjustments**. Some model results were adjusted due to missing data on sectors and sub-sectors, different accounting approaches or too broad regional definitions. The DNE21 + (on country level) does not include the Agricultural, Forestry and Land Use (AFOLU) sector. Therefore, these were supplemented with average estimates from the other global models. Although the POLES model does include AFOLU $CO_2$ emissions, based on estimates from national communications, they were harmonised with those from FAOSTAT[46], as the accounting approaches of the individual countries were not consistent with the other IAMs. The COPPE-COFFEE model does not include F-gas emissions, which were supplemented with average estimates from the other global models. Some national models only cover energy $CO_2$ emissions (China TIMES, China IPAC-AIM V1.0, AIM India, MARKAL India, PRIMES, RU-TIMES) and industrial $CO_2$ emissions and non-$CO_2$ emissions were supplemented with average model estimates from global models.

**Reporting summary**. Further information on research design is available in the Nature Research Reporting Summary linked to this article.

## Data availability

Data reported in Figs. 1–5 and the selection of policies implemented in IAMs can be found in the Source Data. The source data files are also available at [https://doi.org/10.17632/2j7sksfh2h.1]. The list of policies is based on the open source Climate Policy Database. The scenario protocol and the selection of high-impact policies that were included in the protocol are found under Work Package 2 of the deliverables & publications page of the CD-LINKS project. Model results can be found in the open-access CD-LINKS database. Policy relevant data is available in the Global Stocktake tool. CD-LINKS inventory http://www.climatepolicydatabase.org/index.php/CDlinks_policy_inventory; Climate policy database http://climatepolicydatabase.org/index.php/Climate_Policy_Database; Deliverables & publications http://www.cd-links.org/?page_id=620; CD-LINKS database https://db1.ene.iiasa.ac.at/CDLINKSDB/dsd?Action=htmlpage&page=30; Global Stocktake tool https://themasites.pbl.nl/global-stocktake-indicators/.

## Code availability

The code from the 20 integrated assessment models is not available in a publicly shareable version, although several have published open source code, visualisation tools or detailed documentation (see Supplementary Table 10 for details). A model description (see Supplementary Note), and a description of how national climate policies have been implemented (see Supplementary Table 4) is available.

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

## Acknowledgements

We would like to thank the following people for reviewing the CD-LINKS climate policy database: Chenmin He from Energy Research Institute of the National Development and Research Commission, China (NDRC-ERI), Zbigniew Klimont, Nicklas Forsell, Jessica Jewell and Olga Turkovska from International Institute for Applied Systems Analysis (IIASA), Amit Garg from the Public Systems Group at the Indian Institute of Management, India (IIM), Roberta Pierfederici from Institute for Sustainable Development and International Relations (IDDRI), Ucok WR Siagian from Institut Teknologi Bandung, Indonesia (ITB), Jiyong Eom and Cheolhung Cho from Korea Advanced Institute of Science and Technology, Republic of Korea (KAIST), Takeshi Kuramochi from NewClimate Institute (NCI), Junichiro Oda from Research Institute of Innovative Technology for the Earth, Japan (RITE), Aayushi Awasthy and Swapnil Shekhar from The Energy and Resources Institute, India (TERI), Hongjun Zhang from Tsinghua University, China (TU), Nick Macaluso from Environment and Climate Change Canada (EC), Michael Boulle, Hilton Trollipp from Energy Research Centre, South Africa (ERC) and Daniel Buira (Mexico), Vladimir Potachnikov from National Research University Higher School of Economics (Russian Federation). This work is part of a project funded by the European Union's Horizon 2020 Research and Innovation Programme under grant agreement No. 642147 (CD-LINKS), and is supported by European Union's Horizon 2020 Research and Innovation Programme under grant agreement No. 821471 (ENGAGE) and European Union's DG CLIMA and EuropeAid under grant agreement No. 21020701/2017/770447/SER/CLIMA.C.1 EuropeAid/138417/DH/SER/MulitOC (COMMIT). S. F., K. O.: supported by the Environment Research and Technology Development Fund (2-1908 and 2-1702) of the Environmental Restoration and Conservation Agency. J. D., K. K.: the views expressed are purely those of the writer and may not in any circumstances be regarded as stating an official position of the European Commission.

## Author contributions

M.R., D.V. wrote the paper, and all authors contributed to the analysis and article review. Figures were created by H.v.S and M.R.. M.R., H.v.S., D.v.V., M.d.E. and F.U. coordinated the analysis for this paper. The policy inventory and database was created by N.H., G.I., M.R., H.v.S. and D.v.V. The CD-LINKS project was supervised by K.R. and V.K., and advised by J.E. M.H., E.K., G.L., K.R., M.R., H.v.S. and D.v.V. coordinated the global modelling exercise, and C.B., D.H., V.K., E.K., G.L., K.R., R.S., H.v.S., F.U. and D.v.V. coordinated the national modelling exercise. D.v.V. and N.H. supervised the collection of policies, and D.v.V. and M.R. the protocol for model runs. The scenario database was coordinated by D.H. and V.K. Global model runs (incl. documentation) were accomplished by M.H., M.R., H.v.S. (IMAGE), C.B., F.H., E.K., G.L., F.U. (REMIND), S.F., O.F., M.G., V.K. (MESSAGE), L.D., J.E., L.A.R. (WITCH), Z.V., K.F. (GEM-E3), J.D., K.K. (POLES), R.S., P.R. (COPPE-COFFEE), A.K. (BLUES), S.F., K.O. (AIM/CGE, AIM Enduse Japan), K.G. (DNE21+), W.C. (China TIMES), G.I. (GCAM-USA), M.K. (PRIMES), G.S. (RU-TIMES), S.S.V. (AIM India), J.K. (IPAC China), R.M. (MARKAL India).

## Competing interests

The authors declare no competing interests.

## Additional information

Mark Roelfsema[1✉], Heleen L. van Soest[1,2], Mathijs Harmsen[1,2], Detlef P. van Vuuren[1,2], Christoph Bertram[3], Michel den Elzen[2], Niklas Höhne[4,5], Gabriela Iacobuta[4], Volker Krey[6], Elmar Kriegler[3], Gunnar Luderer[3,7], Keywan Riahi[6], Falko Ueckerdt[3], Jacques Després[8], Laurent Drouet[9], Johannes Emmerling[9], Stefan Frank[6], Oliver Fricko[6], Matthew Gidden[6,10], Florian Humpenöder[3], Daniel Huppmann[6], Shinichiro Fujimori[11], Kostas Fragkiadakis[12], Keii Gi[13], Kimon Keramidas[8], Alexandre C. Köberle[14,15], Lara Aleluia Reis[9], Pedro Rochedo[14], Roberto Schaeffer[14], Ken Oshiro[11],

Zoi Vrontsi[12], Wenying Chen[16], Gokul C. Iyer [17], Jae Edmonds [17], Maria Kannavou[12], Kejun Jiang[18], Ritu Mathur[19], George Safonov [20] & Saritha Sudharmma Vishwanathan[21,22]

[1]Copernicus Institute of Sustainable Development, Utrecht University, Princetonlaan 8a, 3584 CB Utrecht, The Netherlands. [2]PBL Netherlands Environmental Assessment Agency, PO Box 30314, 2500 GH The Hague, The Netherlands. [3]Potsdam Institute for Climate Impact Research (PIK), Member of the Leibniz Association, PO Box 601203, 14412 Potsdam, Germany. [4]Environmental Systems Analysis Group, Wageningen University & Research, PO Box 47, 6700 AA Wageningen, The Netherlands. [5]NewClimate Institute, Clever Strasse 13–15, 50668 Cologne, Germany. [6]International Institute for Applied Systems Analysis (IIASA), Schlossplatz 1, 2361 Laxenburg, Austria. [7]Chair of Global Energy Systems, Technische Universität Berlin, Straße des 17. Juni 135, 10623 Berlin, Germany. [8]European Commission, Joint Research Centre, Edificio Expo, C/Inca Garcilaso, 3, 41092 Seville, Spain. [9]RFF-CMCC European Institute on Economics and the Environment (EIEE), Centro Euro-Mediterraneo sui Cambiamenti Climatici, Via Bergognone, 34, 20144 Milan, Italy. [10]Climate Analytics, Ritterstrasse 3, Berlin, Germany. [11]Kyoto University, C1-3, Kyotodaigaku-Katsura, Nishikyo-ku, Kyoto, Japan. [12]E3M-Lab, Institute of Communication and Computer Systems, National Technical University of Athens, Iroon Politechniou Street, 15 773 Zografou Campus, Athens, Greece. [13]Research Institute of Innovative Technology for the Earth, Kyoto 619-0292, Japan. [14]COPPE, Universidade Federal do Rio de Janeiro, PO Box 68565, 21941-914 Rio de Janeiro, RJ, Brazil. [15]Grantham Institute, Imperial College London, Exhibition Road, London SW7 2AZ, UK. [16]Institute of Energy, Environment and Economy, Tsinghua University, 100084 Beijing, China. [17]Joint Global Change Research Institute, Pacific Northwest National Laboratory, 5825 University Research Court, Suite 3500, College Park, MD 20740, USA. [18]Energy Research Institute, National Development and Reform Commission, B1505, Guohong Building, Jia.No.11, Muxidibeili, 100038 Beijing, Xicheng District, China. [19]The Energy & Resources Institute (TERI), India Habitat Center, Lodhi Road, New Delhi-3, India. [20]National Research University Higher School of Economics (HSE), 20 Myasnitskaya street, Moscow, Russian Federation 101000. [21]Indian Institute of Management-Ahmedabad, Public Systems Group, Vastrapur, Ahmedabad, Gujarat, India. [22]National Institute for Environmental Studies, 16-2 Onogawa, Tsukuba, Ibaraki, Japan. ✉email: m.r.roelfsema@uu.nl

