## [Peer Review File · Nature Communications]

Reviewers' comments:

Reviewer #1 (Remarks to the Author):

This paper uses a suite of global and national IAMs to evaluate the effect of national climate policies in the context of the Paris Agreement. The paper finds that national policies and NDCs come up short, and that there's quite a bit of heterogeneity across countries. The authors have put in admirable effort to collect detailed national-level policy and bring them into a large set of IAMs.

Major Comments

1. I'd be very interested in what the carbon prices ended up being for the deeper cut scenarios. We can take this as a least-cost benchmark to lower bound how big of an effort, in dollar terms, will be required to reach the Paris goals.
2. Related to 1: The big hook of the paper is taking national policies seriously, but the gap between national policies and the Paris targets is done with a highly efficient global policy. How much would things change if this carbon price was not a viable option so quickly or at all?
3. The writing could be improved in a lot of places, e.g. line 167 (not included based impact based on), line 188 (global ambition gap and for seven large emitting G20 countries), etc
4. I found Figure 4 a bit confusing in comparing cumulative emissions over a 40 year time frame to a single year's emissions. From what I gather the Figure is indicating that the US even with no new policies is not expected to increase emissions while India is expected to see effectively a doubling of annual emissions, on average?

Minor Comments

1. I'm uncertain what you mean when you write "aspirational goals" in the methods.
2. One thing I was not 100% sure about was whether a policy that was passed by legislature in 2016 but goes into effect in 2018 would be counted as "implemented."

Reviewer #2 (Remarks to the Author):

The paper reports the results of a set of IAMs and national models applied to national climate policies, to see if the goals of the Paris Agreement can be reached. This type of work is highly relevant for informing politicians and citizens about the effectiveness and sufficiency of the Paris Agreement, as well as where additional policy effort is required. The author team includes relevant experts for this kind of work. Several aspects of the study are, though, unclear to me, while certain choices lack a good motivation. As a result, I find it difficult to judge the overall results and their significance.

Several studies have been undertaken in this vein as acknowledged in references 4-6. It should be made clearer early on (even in the abstract) what this study precisely adds to these. For instance, Vrontisi et al. (2018 ERL) also offer a multi-model assessment. Perhaps the main innovation is the full detail of national policies, but this should then be better explained, along with limitations. Surely some aggregation is adopted, and not all possible policies are modelled. Indeed, you say that the climate policy inventory includes "a description of 215 high impact climate and energy policies for the G20 economies that have been implemented before 2017." This suggests a selection was made out of a

wider set of policies: what are/were the criteria for selection of these policies. Just saying “This inventory was evaluated and expanded by national experts” is a too weak basis for a scientific article. The reader will want to see more details about criteria and shortcomings of the selection.

One also wonders why the focus is on policies implemented before 2017. Most of these were likely already planned before the Paris Agreement. This seems a particularly weak aspect of the study. One would expect it to consider (also) policies that are considered or planned as a result of the Paris Agreement goals. Perhaps this is part of the scenarios, but this is not made sufficiently clear from the start on.

Another shortcoming is that the study covers only 65% of global emissions, due to a focus on the seven largest emitting countries. This deserves a good motivation. Have the authors intended to analyse a larger share of emissions or was this impossible for some reason. Some clarification of this seems pertinent. Note in this respect that the selected countries cover only a small part of the wide diversity of NDCs. According to a recent study by King and Van den Bergh (2019, ERL), which identifies four categories of NDCs, only the first and third of these are covered here, whereas the second and fourth, where one would expect the weakest policies, are lacking. This may then arguably result in overly optimistic findings of your study.

It should also be made clear early on why precisely nine global IAMs and seven national models are used. Are the global IAMs used specifically useful to deal with the selected countries (because of relevant geographical disaggregation)? And why use both global and national models. The reader may guess why, but it would be good if this is motivated explicitly.

The five scenarios come out of the blue. They are not sufficiently clearly explained and motivated – neither the need for them, nor their complementary roles. It would anyway be good to have a table comparing the scenarios along various dimensions (policies, time horizon, etc.), which would surely add to clarification. If your purpose is to study implemented policies, why are then scenarios needed? It seems this needs better explanation. From one line mentioning “different levels of climate policy” I suspect it has to do with policies becoming more stringent over time, such as a carbon tax rising over time? But would it not be good then to separate between scenarios that include time patterns of policies as agreed in policy legislation, and scenarios that make assumptions about future policies? It seems these things are now mixed in single scenarios, or at least it is not very clear. Btw, table 1 is not very clear and neither well explained in the text.

I further was confused by the phrase “Finally, the 2 °C and 1.5 °C scenarios are used to explore what is needed after 2020 to keep global warming below 2° C ...”. I thought the study was positive but here it becomes normative. Moreover, we know without any policy detail how much global emissions need to be reduced for a certain temperature target. So why are these scenarios then needed. Or is it because the national emissions reduction is otherwise not uniquely specified? Clearly, more clarification is needed here.

With regard to the results, I would not be interested in details about sectors (transport, buildings, etc.) or decomposition (energy intensity, final energy use, etc.). Instead, I would want to see graphs/tables that allow comparing the results between the scenarios and with the 1.5/2C requirements in terms of emissions reduction. What surprised me is that in discussing the results no separation was made between the global and national models – I would have expected some statements on this distinction.

The title of the section “National carbon budgets up to 2050” is intriguing but also unclear. What are “national carbon budgets” (perhaps there is a need for them, but they were not established in the Paris Agreement)? Perhaps you want to talk about an approximate or reasonable joint carbon budget

for the seven selected countries in your study. This is not clear though. Perhaps the title should simply be changed to "Cumulative emissions up to 2050".

I find the conclusions difficult reading. Too many details which hinder a clear perspective on the main finding I had expected: which % of required emissions reduction for 1.5 and 2C targets, respectively, is achieved by implemented and planned policies; and this might be compared (also percentually) with the NDCs as the maximum achievement.

Detail: "Flexible instruments" is an unclear term to me – what are inflexible instruments? What matters more is effective instruments, which indeed include carbon pricing. But few countries have implemented this instrument with a serious price level. Might be useful to present insights on what specific instruments by themselves achieve. This brings me to another idea: why not present results decomposed for instrument types. Then the reader can learn about which instruments are most contributing to emissions reduction. Now you only say that "The main conclusion is that all countries need to strengthen climate policy." But this is nothing new. You don't provide details on what instruments should be aimed for based on the insights from your study. That seems a missed opportunity. Instead you end unconvincingly with vague notions like "polycentric climate policy pathways" and "policy learning". That's not a strong way to close your article. Instead, try to make clear to politicians which policies will do the job and which not, according to your study's findings.

The abstract can also be improved in my view. I would expect a clear statement on which temperature trajectory current policies jointly are (i.e. translate the emissions gap of 22.4-28.2 GtCO₂eq to a rough temperature scenario). Or at least indicate which percentage of needed emissions reduction (for 2C) is achieved with the policies. Also good to say explicitly whether the policies assessed are implemented or include also proposed ones.

Reviewers' comments:

Reviewer #1 (Remarks to the Author):

This paper uses a suite of global and national IAMs to evaluate the effect of national climate policies in the context of the Paris Agreement. The paper finds that national policies and NDCs come up short, and that there's quite a bit of heterogeneity across countries. The authors have put in admirable effort to collect detailed national-level policy and bring them into a large set of IAMs.

Thank you for your constructive review. We have processed your comments, and provide detail on this below.

Major Comments

1. I'd be very interested in what the carbon prices ended up being for the deeper cut scenarios. We can take this as a least-cost benchmark to lower bound how big of an effort, in dollar terms, will be required to reach the Paris goals.

- We agree that a cost-indicator would be a good addition to the indicators that are currently shown. As marginal prices are quite model-specific, we instead have added mitigation cost/GDP in Figure 3.

2. Related to 1: The big hook of the paper is taking national policies seriously, but the gap between national policies and the Paris targets is done with a highly efficient global policy. How much would things change if this carbon price was not a viable option so quickly or at all?

- This is an interesting question, but also hard to answer as there are many ways of doing this. We have added a sentence describing that other mechanisms beyond cost-effective are possible. For example Kriegler et al (2018) and Roelfsema et al (2019) introduce the concept of 'good practice policies' that scale up existing policies to the global level, which can be seen. Other methods are introducing 'effort sharing' mechanisms that look at different ways of sharing emission reductions in a "fair" way. These were used to compare with our results (see Supplementary Information). In the end there are many (other) ways of defining mechanisms for climate policy implementation. In this paper, we specifically want to focus on the current national policies as formulated now. It would be interesting for further research though, to define different criteria (also beyond effort sharing) to assess the collective impact on global GHG emissions.

3. The writing could be improved in a lot of places, e.g. line 167 (not included based impact based on), line 188 (global ambition gap and for seven large emitting G20 countries), etc

An English editor has edited the new version of the text

4. I found Figure 4 a bit confusing in comparing cumulative emissions over a 40 year time frame to a single year's emissions. From what I gather the Figure is indicating that the US even with no new policies is not expected to increase emissions while India is expected to see effectively a doubling of annual emissions, on average?

Your interpretation is correct. We use this indicator as it allows us to compare the cumulative emissions of countries given the differences in absolute values. We have added the definition "the number of years you can emit at 2010 emission levels while staying below the total cumulative

emissions of the next 40 years. A value of 40 indicates that, on average, the emission level will remain constant”.

Minor Comments

1. I'm uncertain what you mean when you write "aspirational goals" in the methods.

- These are goals that have not been secured by legislation or executive orders. We have added this in brackets to the text

2. One thing I was not 100% sure about was whether a policy that was passed by legislature in 2016 but goes into effect in 2018 would be counted as "implemented."

- Yes, we assume that as soon as it is passed by legislature, it is implemented.

Reviewer #2 (Remarks to the Author):

The paper reports the results of a set of IAMs and national models applied to national climate policies, to see if the goals of the Paris Agreement can be reached. This type of work is highly relevant for informing politicians and citizens about the effectiveness and sufficiency of the Paris Agreement, as well as where additional policy effort is required. The author team includes relevant experts for this kind of work. Several aspects of the study are, though, unclear to me, while certain choices lack a good motivation. As a result, I find it difficult to judge the overall results and their significance.

Thank you for your constructive feedback. We have improved the article by making certain topics more clear, and add better motivations. Please, find below our description of improvements based on your comments.

Several studies have been undertaken in this vein as acknowledged in references 4-6. It should be made clearer early on (even in the abstract) what this study precisely adds to these. For instance, Vrontisi et al. (2018 ERL) also offer a multi-model assessment. Perhaps the main innovation is the full detail of national policies, but this should then be better explained, along with limitations. Surely some aggregation is adopted, and not all possible policies are modelled. Indeed, you say that the climate policy inventory includes “a description of 215 high impact climate and energy policies for the G20 economies that have been implemented before 2017.” This suggests a selection was made out of a wider set of policies: what are/were the criteria for selection of these policies. Just saying “This inventory was evaluated and expanded by national experts” is a too weak basis for a scientific article. The reader will want to see more details about criteria and shortcomings of the selection.

- We have improved the description on what our paper adds, also to Vrontisi.
 - Vrontisi et al (2018) only includes a NDC scenario, in addition to a reference scenario that includes Copenhagen pledges. Our main addition is to include a detailed representation of current implemented policies, going significantly beyond the work

of Vrontisi et al. We have made this more clear and a summary of this was added to the abstract.

- We have made a selection of the most important policies, and the main limitation is that in some cases policies needed to be translated into the representation of specific IAMs (e.g. some models can directly implement CAFE standard, but others simply have to include this as efficiency improvement in transport). Moreover, some models only include G20 countries as part of a larger model regions. Although we had already some description in the text, we have reworded it and put the text directly after the sentence on what this paper adds to literature.
- We have expanded the description of how policies were selected in the text. First a large database was setup that includes all policies that could be found, from either the NDCs (the often mention specific policies that are already implemented), literature and from a survey to country experts. Then, for each G20 country ten policies were selected from the database in consultation with policy experts of each country 1) that have expected high impact on GHG emissions, based on literature and the opinion of our experts, 2) those policies are adopted by government through legislation or executive orders, and 3) no evidence exists of large barriers to implementation. This process was done together with national experts, and consisted of two rounds.

One also wonders why the focus is on policies implemented before 2017. Most of these were likely already planned before the Paris Agreement. This seems a particularly weak aspect of the study. One would expect it to consider (also) policies that are considered or planned as a result of the Paris Agreement goals. Perhaps this is part of the scenarios, but this is not made sufficiently clear from the start on.

- As most NDCs were submitted some months before the Paris Agreement, countries already had many policies in place by 2017. The paper 'National climate change mitigation legislation, strategy and targets: a global update' (Iacobuta et al, 2018) shows that in the years before 2015 many new policies were introduced, while this levelled off significantly between 2015 and 2017.
- In addition, policy implementation is an ongoing process and one needs to determine a cut-off data. It takes time to collect policies and implement and change this for a large set of integrated assessment models.

Another shortcoming is that the study covers only 65% of global emissions, due to a focus on the seven largest emitting countries. This deserves a good motivation. Have the authors intended to analyse a larger share of emissions or was this impossible for some reason. Some clarification of this seems pertinent. Note in this respect that the selected countries cover only a small part of the wide diversity of NDCs. According to a recent study by King and Van den Bergh (2019, ERL), which identifies four categories of NDCs, only the first and third of these are covered here, whereas the second and fourth, where one would expect the weakest policies, are lacking. This may then arguably result in overly optimistic findings of your study.

- This could have been unclear in the text. Although we present results for seven large countries, we have included climate policies for all G20 countries, representing 75% of total GHG emissions in 2010. This was done as selection of the policies and implementation into models is very time consuming, and because of comparability across the models (some IAMs have a limited regional coverage). Therefore, we present only the largest (G20) countries individually. This limitation has been added to the text.

- Although most countries have submitted NDC, there are not many (concrete) domestic policies in place for other countries (e.g. many African countries) that will help achieve the pledges. Based on the Climate Action tracker we found the countries with climate policies, but which were not included in our assessment. They represent somewhat less than 5% of global 2010 emissions (see Supplementary Information).
- We cannot see that the current policies scenario is overly optimistic. We therefore assume you are referring to the NDC scenario? But the study is exactly about identifying that the NDCs ambitions have not been translated in implemented policies yet (legislation in place).
- In the NDC scenario, we cover the first, second, and third category from King and Van den Bergh (2019, ERL) for G20 countries. For the fourth category we assume business-as-usual (SSP2). Actually, the NDC for China and India go beyond the 'emissions intensity reductions' category, as they also pledge to achieve a non-fossil target and a forestry target, which are also modelled in our scenario. All assumptions for the NDC scenario can be found in the Supplementary Excel file that contains both policies that are included in the national policies scenario, as the NDC assumptions (including LULUCF). The description in the text was extended.

It should also be made clear early on why precisely nine global IAMs and seven national models are used. Are the global IAMs used specifically useful to deal with the selected countries (because of relevant geographical disaggregation)? And why use both global and national models. The reader may guess why, but it would be good if this is motivated explicitly.

- A description of how global and national models were selected was added to the text.
 - Global models
 - Large selection of models that submitted scenarios with policies to the IPCC 1.5C report
 - Range of different model types (scenario/optimisation, general/partial equilibrium) for a robust assessment
 - Can deal with (most) of the seven large countries, and at least able to implement policies on an aggregated level for the selected G20 countries
 - National models: we have investigated which national G20 models were available, that were able to show the impact of national climate policies. These models were found in China, USA, India, EU, Japan, Brazil and Russia. This was added to the text.
 - To evaluate the coherence of the global emission pathways, we compared these to those of the national models. This was only done for the *new policies scenario* and *NDC scenario*, as designing 2 °C/1.5 °C scenarios in the same way as was done for global models (global cost-effective implementation) is not straightforward. This is beyond the scope of this paper.

The five scenarios come out of the blue. They are not sufficiently clearly explained and motivated – neither the need for them, nor their complementary roles. It would anyway be good to have a table comparing the scenarios along various dimensions (policies, time horizon, etc.), which would surely add to clarification. If your purpose is to study implemented policies, why are then scenarios needed? It seems this needs better explanation. From one line mentioning “different levels of climate policy” I suspect it has to do with policies becoming more stringent over time, such as a carbon tax rising over time? But would it not be good then to separate between scenarios that include time patterns of policies as agreed in policy legislation, and scenarios that make assumptions about future policies? It seems these things are now mixed in single scenarios, or at least it is not very clear. Btw, table 1 is not very clear and neither well explained in the text.

- Note that national climate policies were not implemented through carbon taxes, but by replicating the policy target underlying the policy instrument. This is the main improvement of this study, which is a better representation of policies.
- A table was added showing the different scenario assumptions and the reasoning behind the scenarios, which is now better explained in the text.
- However, the selection is very logical given the purpose of the article. In the Paris Agreement, countries promised to formulate climate policy in order to stay well-below 2 degrees C and preferably 1.5 degree. For this, countries promised contributions (NDCs). Only part of the promises, however, has yet been translated into legislative policies. In the paper, we use the corresponding scenarios to show the differences between these ambitions and policies – both in terms of an ambition gap (difference with 1.5/2 deg C level) and implementation gap (not sufficiently policies implemented). We finally compare all scenarios with a scenario without policies.

I further was confused by the phrase “Finally, the 2 °C and 1.5 °C scenarios are used to explore what is needed after 2020 to keep global warming below 2° C ...”. I thought the study was positive but here it becomes normative. Moreover, we know without any policy detail how much global emissions need to be reduced for a certain temperature target. So why are these scenarios then needed. Or is it because the national emissions reduction is otherwise not uniquely specified? Clearly, more clarification is needed here.

We have changed the wording, and do not use the word ‘needed’ anymore. As explained to the previous comment, in the Paris Agreement countries set as overall objective to reach the 1.5 / 2 degree target. This study evaluates implemented policies and see whether they are (already) consistent with the overall ambition. The 2/1.5 scenario are therefore used as comparison. We disagree that the emission levels for the 2/1.5 targets can be directly derived from the targets: choices can be made between emission reductions in time and across gases. We used the models to identify cost-optimal mitigation strategies starting from 2020. While one may argue that there are alternative 2/1.5 strategies, we show that the reductions in the current policies scenario are still so far away from those in the 2/1.5 case, that one can confidently start to suggest how to strengthen current policies and NDCs. We have reworded the sentence to make this more clear – but also already on the basis of your previous comment.

With regard to the results, I would not be interested in details about sectors (transport, buildings, etc.) or decomposition (energy intensity, final energy use, etc.). Instead, I would want to see graphs/tables that allow comparing the results between the scenarios and with the 1.5/2C requirements in terms of emissions reduction. What surprised me is that in discussing the results no separation was made between the global and national models – I would have expected some statements on this distinction.

- We have added two tables with detailed results on the impact on global and national GHG emissions
- But in addition, we think that more insights into sector results and indicators beyond GHG emissions could help countries with identifying areas to enhance their policy implementation.
- The national models are added to the analysis to give some confidence that the emissions reductions shown can be achieved, especially for the current policy scenarios. The results show that reductions for this scenarios are indeed comparable. This remark is added more clearly in the text.

The title of the section “National carbon budgets up to 2050” is intriguing but also unclear. What are “national carbon budgets” (perhaps there is a need for them, but they were not established in the Paris Agreement)? Perhaps you want to talk about an approximate or reasonable joint carbon budget for the seven selected countries in your study. This is not clear though. Perhaps the title should simply be changed to “Cumulative emissions up to 2050”.

- Using the word budget is indeed somewhat confusing. We changed the title, and removed the word ‘budget’/

I find the conclusions difficult reading. Too many details which hinder a clear perspective on the main finding I had expected: which % of required emissions reduction for 1.5 and 2C targets, respectively, is achieved by implemented and planned policies; and this might be compared (also percentually) with the NDCs as the maximum achievement.

- We have changed the conclusions, and now start with the impact on GHG emissions by implementing national policies, and give the emissions gap with 2 °C and 1.5 °C scenarios. In addition, provide how much smaller this gap is, if conditional NDCs would be fully implemented.

Detail: “Flexible instruments” is an unclear term to me – what are inflexible instruments? What matters more is effective instruments, which indeed include carbon pricing. But few countries have implemented this instrument with a serious price level. Might be useful to present insights on what specific instruments by themselves achieve. This brings me to another idea: why not present results decomposed for instrument types. Then the reader can learn about which instruments are most contributing to emissions reduction. Now you only say that “The main conclusion is that all countries need to strengthen climate policy.” But this is nothing new. You don’t provide details on what instruments should be aimed for based on the insights from your study. That seems a missed opportunity. Instead you end unconvincingly with vague notions like “polycentric climate policy pathways” and “policy learning”. That’s not a strong way to close your article. Instead, try to make clear to politicians which policies will do the job and which not, according to your study’s findings.

- The word flexible instruments is used in climate policy making to refer to emission trading and joint-implementation. But as it is jargon we have now changed the wording for ‘flexible instruments’
- Although your idea on the impact of individual policies is definitely interesting, and was also discussed already during our research. But is very difficult to implement in a multi-model study. As here, we wanted to focus mostly on evaluating for the first time the current status of policies, we have decided to benefit from the additional robustness provided by using multiple models. In follow-up work, it is possible to evaluate individual policy types, as this is still quite some work as one would need to identify the interaction (and impact) between different instruments.
- We have changed the final remarks in the conclusions section to address policy makers and make it more concrete.

The abstract can also be improved in my view. I would expect a clear statement on which temperature trajectory current policies jointly are (i.e. translate the emissions gap of 22.4-28.2 GtCO₂eq to a rough temperature scenario). Or at least indicate which percentage of needed emissions reduction (for 2C) is achieved with the policies. Also good to say explicitly whether the

policies assessed are implemented or include also proposed ones.

- We have changed the abstract
- Adding temperature statements for the end of the century would be rather speculative, as our paper assess climate policies that are mostly implemented before 2030. We do extend our policy scenario with 20 years to 2050 by assuming equal reductions relative to the no new policies scenario, but doing this for a total of 70 years would not be sensible.
- We have added the percentage of emission reductions between the No new policies scenario and the 2C/1.5C to that is achieved with policies to the text and abstract.

Reviewers' comments:

Reviewer #1 (Remarks to the Author):

Very nice paper.

Reviewer #2 (Remarks to the Author):

I thank the authors for the clear responses to my comments, questions and suggestions.

While the paper has been improved in many respects, I still feel it is difficult to fully understand the (rationale of) the approach and the policy lessons derived.

One has to read a lot of text in the fairly long introduction before one can understand the overall approach. It would be good to summarize all main elements of the approach in a coherent way in a short paragraph early on.

The final section needs more work still in my view:

Regarding policy advice, given that we are so far removed from the necessary policies, I would expect a more critical stance on policy mixes with many weak instruments which evidently don't add up to sufficient regulatory power, and also on sector policies which lead to inconsistent implicit carbon prices, and hence too cost-ineffective solutions and carbon leakage between sectors within countries.

However, no clear policy lessons are offered, which I find a bit disappointing. Without this, you offer just one more (of many) studies which clarifies what we already know: the Paris Agreement falls (enormously) short of its aims.

I wonder if the models account sufficiently for carbon leakage, rebound and oil market responses (green paradox). If not, then the results are probably even overly optimistic. A note on this would be fair.

"Some countries are close to achieving NDCs, but a large gap with 2 °C or 1.5 °C 336 pathways then remains." Odd sentence. Perhaps more informative to say how many countries are close to, and how many far off from, achieving their NDC.

Regarding title and abstract:

I feel the title is still not capturing well the originality and approach of the paper. Good to avoid repeating terms in the title ("climate policies" appears twice; "climate" appears 3x).

The abstract is amenable for improvement. Phrase "for the first time, a multi-model analysis" – why is "multi-model" so relevant? Not motivated. "covering 11 to 15.4% (median) of the emissions gap" is not very clear as emissions gap is not defined. Is the conclusion that more than 80% of required emission reduction will not be covered by planned policies? If so, say more clearly and explicitly. But then the 36-45% are not clear. I would anyway delete the final part of the first sentence ("resulting in ... pathways") and focus on the second one. Now there are too many seemingly inconsistent numbers. "Russian Federation final energy intensity" too long term and possessive "s" missing after "Federation".

Reviewer #1 (Remarks to the Author):

Very nice paper.

Reviewer #2 (Remarks to the Author):

I thank the authors for the clear responses to my comments, questions and suggestions.

While the paper has been improved in many respects, I still feel it is difficult to fully understand the (rationale of) the approach and the policy lessons derived.

Many thanks for the critical review and suggestions. Please find below our responses, including how we have processed them in the new version of the article.

One has to read a lot of text in the fairly long introduction before one can understand the overall approach. It would be good to summarize all main elements of the approach in a coherent way in a short paragraph early on.

Many thanks for the critical review and suggestions. Please find below our responses, including how we have processed them in the new version of the article.

We have rewritten, restructured and shortened the introduction without changing the message and without leaving out important background information. It now presents the aim of the paper already in the second section. In addition, repeated statements in the text were removed. Some more methodological details were moved to the methods section.

The final section needs more work still in my view:

Regarding policy advice, given that we are so far removed from the necessary policies, I would expect a more **critical stance on policy mixes** with many weak instruments which evidently don't add up to sufficient regulatory power, and also on sector policies which lead to inconsistent implicit carbon prices, and hence too cost-ineffective solutions and carbon leakage between sectors within countries. However, **no clear policy lessons are offered**, which I find a bit disappointing. Without this, you offer just one more (of many) studies which clarifies what we already know: the Paris Agreement falls (enormously) short of its aims.

Thanks for the suggestion. We have not analysed the policy mixes in detail (apart from aggregate impact), however, discussing the ineffective policy mixes is an interesting angle. We have added this to the discussion, suggesting that a careful redesign of the current policy mixes, considering effectiveness and other criteria, could avoid carbon leakage, avoid stranded assets and increase regulatory power of governments. This is accompanied with literature references to this topic.

I wonder if the models account sufficiently for carbon leakage, rebound and oil market responses (green paradox). If not, then the results are probably even overly optimistic. A note on this would be fair.

This is different for different models, especially between economic models and the more technological rich non-economic models. All models do have a dynamic pricing, therefore local climate policy has impact on fossil fuel prices leading to changes in climate policy in other regions. On the other hand, underestimating the impact of technological learning (e.g. as was visible with the German feed-in-tariff) would lead to the opposite effect. A note on this topic (see paragraph Integrated Assessment Models (IAMs)) is added to the Methods section.

“Some countries are close to achieving NDCs, but a large gap with 2 °C or 1.5 °C 336 pathways then remains.” Odd sentence. Perhaps more informative to say how many countries are close to, and how many far off from, achieving their NDC.

Sentence has been removed and replaced by other sentence that was already in the text on countries' achievement.

Regarding title and abstract:

I feel the title is still not capturing well the originality and approach of the paper. Good to avoid repeating terms in the title (“climate policies” appears twice; “climate” appears 3x).

The title was changed to **‘Taking stock of national climate policies: the Paris Agreement needs to speed up implementation and scale up ambition’**. We kept the concept of ‘stock taking’, as the paper addresses this process in the Paris Agreement. However, we have added the main conclusion, that in addition to ambition gaps, countries also need to close the implementation gaps.

The abstract is amenable for improvement. Phrase “for the first time, a multi-model analysis” – why is “multi-model” so relevant? Not motivated. “covering 11 to 15.4% (median) of the emissions gap” is not very clear as emissions gap is not defined. Is the conclusion that more than 80% of required emission reduction will not be covered by planned policies? If so, say more clearly and explicitly. But then the 36-45% are not clear. I would anyway delete the final part of the first sentence (“resulting in ... pathways”) and focus on the second one. Now there are too many seemingly inconsistent numbers. “Russian Federation final energy intensity” too long term and possessive “s” missing after “Federation”.

The abstract has changed significantly to meet the 150 words limit, and to address your comments. An explanation of why multi-model is relevant is added at the end of the first paragraph in the Introduction. The sentences on how much is covered are deleted, as is the final part of the first sentence.

REVIEWERS' COMMENTS:

Reviewer #2 (Remarks to the Author):

I am happy with the changes made. Everything is now clear.